# A Contemporary Review of Robotic Resection for Hepatocellular Carcinoma

**DOI:** 10.3390/cancers16223806

**Published:** 2024-11-12

**Authors:** William A. Preston, Nina R. Spitofsky, Adam S. Bodzin

**Affiliations:** Department of Surgery, Division of Transplantation, Sidney Kimmel Medical College, Thomas Jefferson University, 1015 Walnut Street, Curtis Building, Suite 613, Philadelphia, PA 19107, USA; william.preston@jefferson.edu (W.A.P.); nina.spitofsky@jefferson.edu (N.R.S.)

**Keywords:** hepatocellular carcinoma (HCC), robotic surgery, outcomes

## Abstract

Minimally invasive surgery has been applied across subspecialties with comparable long-term and oncological outcomes to open surgery, though with decreased morbidity and length of stay. Robotic surgery, in particular, has garnered interest within the liver surgery community because of improved dexterity and 3-dimensional vision, which may aid in the resection of lesions in delicate anatomical spaces. Given the complexity of resections and the often-underlying diseased liver in which they reside, hepatocellular carcinoma is one such lesion that may significantly benefit from this precise resection modality. This study aimed to provide a comprehensive review of contemporary literature comparing robotic hepatectomy to laparoscopic/open hepatectomy in the context of hepatocellular carcinoma.

## 1. Introduction

Hepatocellular carcinoma (HCC) is the most common primary liver malignancy (~90% of cases) and carries a poor prognosis (5-year survival < 20%) [1,2]. Most HCCs arise amid underlying liver disease and are thus associated with conditions such as alcohol consumption, nonalcoholic steatohepatitis (NASH), hepatitis-B, and hepatitis-C, wherein hepatic inflammation results in fibrosis and abnormal hepatocyte regeneration [3,4]. Pursuing is an accumulation of genetic/epigenetic alterations that can result in dysplastic nodules, which have malignant potential [5].

Screening programs for HCC in high-risk patients (i.e., those with cirrhosis, high risk hepatitis-B) are now abundant and are comprised of ultrasonography and alfa-fetoprotein measurement [6]. Diagnosis is typically made utilizing cross-sectional imaging, with findings of hyperenhancement on arterial phase (secondary to angiogenesis), followed by washout on portal venous phase (secondary to hypo-enhancement relative to surrounding non-neoplastic parenchyma) [7]. Because of increased screening availability, better characterization of high-risk patients, and improved diagnostics, HCC is being identified at early stages amendable to transplantation, resection, and/or locoregional therapy. Still, many HCC patients do not meet criteria for transplantation, and those who do have limited access given recent allocation changes (i.e., MMaT-3 [median model for end stage liver disease at transplant policy]) and newer technology; hence, these considerations highlighting the important role of resection [8,9].

Hepatectomy for HCC is unique in that most patients have underlying cirrhosis, often necessitating a more robust future liver remnant than would other lesions arising amongst normal liver parenchyma [10]. Additionally, post-resection recurrence is common, owing to portal/hepatic venous spread, de novo carcinogenesis in the cirrhotic liver, and high-risk biological factors, including lymphovascular invasion and multifocality [11,12,13]. These factors emphasize the delicate balance required to achieve optimal outcomes while maintaining hepatic function and underscore the complexity of procedural selection and intraoperative decision making (i.e., resection vs. locoregional therapy; anatomical vs. non-anatomic resection; margin width). Yet, an increasingly prevalent discussion in the HCC literature pertains to minimally invasive liver resection—namely, robotic liver resection.

Indeed, robotic hepatectomy is an appealing topic in complex liver surgery, as improved articulation, tremor reduction, and magnified 3-dimensional visuals all may aid in safe resection, while limiting the risk of open conversion compared to traditional laparoscopic surgery [14]. A multicenter pilot study by Choi et al. demonstrated that experienced surgeons in open/laparoscopic liver surgery can safely perform major robotic hepatic resections [15]. With the advent of robotic/laparoscopic hepatectomy methods and an increasing number of surgeons capable of performing such procedures, a review and comparison of different surgical approaches is warranted. This review aims to critically review contemporary literature on robotic surgery for HCC (RS) as compared with laparoscopic (LS) and open surgery (OS).

## 2. Methods and Materials

### 2.1. Literature Search

A literature search was performed by two of the authors (W.A.P. and N.R.S.) on https://pubmed.ncbi.nlm.nih.gov/ (Last accessed 30 April 2024). We limited our search to studies between 1/2018–4/2024, including only studies focusing on human subjects which were written in the English language. The following MESH search headings were used: (“Carcinoma, Hepatocellular” [Mesh]) AND (“Robotic Surgical Procedures” [Mesh]). The last search was run on 30 April 2024.

### 2.2. Study Selection

The same two authors screened titles and abstracts of identified studies that met the inclusion criteria above for appropriateness. All relevant studies comparing outcomes between robotic, open, and/or laparoscopic resections for HCC were included, as were those describing measurable outcomes. We excluded case reports, meta-analyses, studies that did not specifically differentiate RS from LS, and studies that were not exclusive to HCC.

## 3. Results

The literature search yielded 74 articles, 14 of which were included for discussion. Ten studies were including comparing RS to LS/OS (n = 943 RS, n = 1678 LS/OS); 3 of these studies compared RS to LS (Table 1, n = 211 RS, n = 505 LS); 7 studies compared RS to OS (Table 2, n = 732 RS, n = 1117 OS). Finally, 1 study compared RS to microwave ablation (Table 3), 1 discussed RS margins, 1 discussed the results of RS in metabolic syndrome, and 1 discussed RS and “huge” (≥10 cm) HCC. 

### 3.1. Robotic vs. Laparoscopic Surgery for HCC

Duong et al. investigated data from the National Cancer Database (a United States of American [USA] cancer database sponsored by the American College of Surgeons and American Cancer Society), including Stage I patients with HCC (American Joint Committee on Cancer Staging System 7th edition) diagnosed between 2010–2015 [16]. Propensity score matching (PSM) was performed, adjusting for year of diagnosis, demographics, comorbidities, tumor characteristics, and additional treatment (chemotherapy/radiation), resulting in 123 RS and 369 LS. Compared to LS, a greater number of RS patients had lymph node tissue retrieved for examination (6.5% vs. 2.0%, *p* < 0.01). Short-term mortality (both 30-day [0.8% RS vs. 1.6% LS] and 90-day [0.8% RS vs. 3.3% LS]) was similar between groups; however, RS was associated with an approximately 40% decreased risk of mortality compared to LS (HR 0.59 [0.39–0.90]), as 5-year survival was 63% for RS vs. 45% for LS (median follow-up time ~28 months for both groups).

With advancements in 3-dimensional laparoscopy, a need for comparison to RS in the context of HCC was necessary. Lim et al. led a European tri-institutional comparative study of 93 consecutive HCC patients undergoing RS (n = 44) vs. 3-dimensional LS (n = 49) major/minor hepatectomy between 2011–2017 [17]. Patient demographics, degree of cirrhosis, and tumor characteristics were similar across groups. Interestingly, operative time was similar between groups (252 min RS vs. 269 min LS); however, stratification by resection extent demonstrated significantly longer operative times for RS major hepatectomy, compared to LS (509 min vs. 394 min, *p* = 0.05). Compared to LS, RS patients were less likely to undergo Pringle maneuver (20% vs. 65%, *p* < 0.0001) and more likely to undergo intraoperative transfusion (14% vs. 2%, *p* = 0.03). Hospital stay was similar between groups (9 days RS vs. 7 days LS, *p* = 0.27), as were rates of post-operative liver failure, biliary complications, and ascites (all <5% in both groups). No postoperative mortalities or readmissions were observed in this cohort and major morbidity rate (2% RS vs. 4% LS), margin width, and rate of R0 resection were comparable between groups. Three-year overall survival (91% RS vs. 82% LS, *p* = 0.16) and 3-year recurrence-free survival (48% RS vs. 24% LS, *p* = 0.18) were similar between groups.

Liu et al. published data from a single institution on 131 patients undergoing major hepatectomy (≥3 consecutive Couinaud segments) for HCC between 2017–2022 (n = 44 RS, n = 87 LS) [18]. Similar baseline characteristics were present between groups, including demographics, tumor size, presence of cirrhosis/steatosis, antiretroviral therapy, hepatitis-B serostatus, and operation type (i.e., right [or extended right] hemihepatectomy, left [or extended left] hemihepatectomy, central bisectionectomy). RS was associated with longer operative time (255.5 min RS vs. 206.8 min LS, *p* < 0.001) while LS was associated with higher estimated blood loss (118.9 mL RS vs. 197 mL LS, *p* = 0.002). No differences were appreciated in conversion to OS (4.5% RS vs. 5.7% LS), length of hospital stay (10.2 days RS vs. 10.3 days LS), post-operative hemorrhage (2.3% RS vs. 2.3% LS), liver failure (2.3% RS vs. 3.4% LS), 30-day mortality (0 both groups), or 90-day mortality (0 both groups). Additionally, no R1 or R2 resection occurred in either group.

### 3.2. Robotic vs. Open Surgery for HCC

In a single-center Taiwanese study by Wang et al. in 2018, outcomes of 63 RS were compared to 177 OS for HCC [19]. Of note, RS was offered to patients with small (<5 cm), peripheral tumors in segments II-VI who did not undergo previous foregut surgery. Aside from tumor size, which was on average smaller in the RS group (3.1 cm RS vs. 3.6 cm OS, *p* = 0.050), demographic and clinical characteristics, including stage, Child-Pugh score, tumor location and number, platelet count, albumin, and treatment with antiretroviral therapy, were similar between groups. The RS group was more likely to undergo non-anatomical wedge resection (67% RS vs. 45% OS, *p* = 0.012) and was associated with longer operative time (296 min RS vs. 182 min OS, *p* = 0.032) and shorter hospital length of stay by almost 2 days (6.21 days RS vs. 8.18 days OS, *p* = 0.001). On the contrary, resection margins (5.9 mm RS vs. 6.4 mm OS), Clavien-Dindo ≥ 3 post-operative complications rate (11.1% RS vs. 15.3% OS), HCC recurrence rate (27% RS vs. 37.3% OS), and time to HCC recurrence (585 days RS vs. 540 days OS) were statistically similar between the groups. Finally, 3-year post-resection survival and post-resection disease-free survival were, respectively, 97.7% RS vs. 92.3% OS (*p* = 0.137) and 71.9% RS vs. 61.6% OS (*p* = 0.325) with no surgical mortality occurring in either group. Finally, while the authors note that resection modality (RS vs. OS) was not a risk factor for HCC recurrence, Child-Pugh score, satellite nodules, and Ishak score 6 (a histological grading marker of chronic hepatitis/fibrosis) all were predictive of recurrence [20].

Pesi et al. published their experience on 54 patients with HCCs < 7 cm without macrovascular invasion who underwent resection (n = 23 RS, n = 31 OS) at 2 European centers [21]. Baseline characteristics, including demographics, comorbidities, cirrhosis and etiology, and Child-Pugh score were statistically comparable between groups. Moreover, segmental lesion location and resection extent were comparable between groups, with most being classified as minor (defined as <3 Couinad segments; 74% RS vs. 84% OS). OS patients had a higher median estimated blood loss (186 mL RS vs. 364 mL OS, *p* = 0.003). Interestingly, operative time was not different between groups (230 min RS vs. 227 min OS, *p* = 0.88), with comparable complication rates (overall 17% RS vs. 23% OS; major [grade 3 or higher]: 9% RS vs. 3% OS, *p* = 0.605). OS vs. RS was not a significant predictor of decreased survival on univariable analysis; however, increased number of lesions (HR 2.39 IQR [1.21–4.70]), and vascular invasion (HR 20.5 [3.1–136.4]) were predictors of poor overall survival on multivariable analysis. Three-year overall survival was 87% RS vs. 78% OS (*p* = 0.203); whereas 3-year disease-free survival was 54% RS vs. 37% OS (*p* = 0.59).

Di Benedetto et al. published a PSM analysis on RS vs. OS for HCC [22]. Including patients from 5 different hospitals across Europe and USA, covariates utilized in PSM included age, body mass index (BMI), Charlson Comorbidity Index, platelet count, IWATE score (an estimate of resection difficulty), and ERASL-pre score (an estimate of early recurrence probability) [23,24]. This resulted in 106 RS and 106 OS patients. Like Wang et al., operative time was longer with RS (295 min RS vs. 200 min OS, *p* < 0.001) and RS was associated with shorter length of stay (4 days RS vs. 10 days OS, *p* < 0.001). Pringle maneuver was performed at a higher rate with RS (13% RS vs. 1% OS, *p* < 0.001), but estimated blood loss was higher (200 mL RS vs. 100 mL OS, *p* < 0.001), despite no differences in transfusion. Nonetheless, it is worth noting that 37.7% of OS were wedge resections, compared to only 17.9% of RS. OS was associated with higher rates of Clavien-Dindo ≥3 complications (2.8% RS vs. 11.3% OS, *p* = 0.029) and post-hepatectomy liver failure (7.5% vs. 28.3%, *p* < 0.001). Similar 90-day post-resection survivals were demonstrated between groups (99.1% RS vs. 97.1% OS, *p* = 0.33), and when resection modality (RS vs. OS) was analyzed for association with survival on multivariable analysis, it was not a significant predictor (HR, 0.52 [0.23–1.16]).

To investigate the impact of RS on the elderly population with HCC, Zhang et al. conducted a Chinese single center, retrospective, PSM analysis from 2010–2020, including patients with HCC, Child-Pugh A/B7 cirrhosis, and R0 resection in patients at least 65 years old [25]. Following PSM, there were 100 patients in the RS group and 178 patients in the OS group. Baseline characteristics, including demographics, HCC etiology, cirrhosis/severity, tumor characteristics, and type of hepatectomy (minor [<3 Couinad segments] vs. major [≥3 Couinad segments]) were similar between groups. Approximately 63% of resections in both groups were classified as minor. The RS group had a lower estimated blood loss (150 mL RS vs. 200 mL OS, *p* = 0.002), though without differences in transfusion rates, lower rate of post-operative complications (Clavien-Dindo, 7% vs. 17.4%, *p* = 0.015), and shorter length of stay (6 days vs. 9 days, *p* < 0.001). Ninety-day mortality was only 1% for each group (*p* = 0.999). Similar rates of 5-year overall survival (43.0% RS vs. 48.8% OS, *p* = 0.722) and recurrence-free survival (20.9% RS vs. 24.6% OS, *p* = 0.982) were noted.

Zhu et al. conducted a propensity score matched analysis comparing RS, LS, and OS for patients with early (Barcelona Clinical Cancer Stage 0-A) HCC [26]. Patients were excluded if they received previous treatment for HCC. All surgeons were proficient in OS and had passed learning curves for RS and LS. Following PSM, 56 patients were included in each group for analysis. RS and LS were associated with longer operative time (220 RS vs. 210 LS vs. 158 OS, *p* < 0.001) and Pringle time (29 min RS vs. 26 min LS vs. 12 min OS, *p* < 0.001). Hospital length of stay was less for RS and LS groups (8 days, 6 days, respectively) compared with OS (12 days, *p* < 0.001). Complication rates were not significantly different between groups. Five-year overall survival (74.4% RS, 76.8% LS, 78.6% OS, *p* = 0.90) did not differ between groups; nor did disease-free survival (51.8% RS, 51.3% LS, 57.9% OS, *p* = 0.64).

Zhang et al. investigated short/long-term outcomes after RS vs. OS for large HCC (≥5 cm), including a subgroup of patients with “huge” HCC (diameter ≥10 cm) between 2010–2020 across 8 Chinese institutions [27]. Patients were excluded if robotic resection required a conversion to laparotomy. Surgeons participating in this study all had significant experience with robotic hepatectomy, defined as having performed ≥30 cases and having passed a set learning curve. PSM yielded 280 RS and 465 OS patients with comparable baseline features, including demographics, presence of cirrhosis, viral serologies, pertinent hepatic labs, varices, tumor size, number of tumors, and macrovascular invasion. No differences were appreciated with respect to Pringle time (26 min RS vs. 21 min OS, *p* = 0.380), though the Pringle maneuver was performed more frequently with OS (70% RS vs. 80% OS, *p* < 0.001). In contrast to the aforementioned studies, shorter median operative time (181 min RS vs. 201 min OS, *p* < 0.001) and lower blood loss (200 mL RS vs. 400 mL OS, *p* < 0.001) were observed in the RS group. RS also evidenced shorter postoperative hospital stay (6 days RS vs. 9 days OS, *p* < 0.001) and reduced major complications (i.e., Clavien-Dindo ≥3; 2% RS vs. 6% OS, *p* = 0.014). Surgical approach had no significant effect on long-term outcomes, as median overall survival (68.9 months RS vs. 64.4 months OS, *p* = 0.475) and median recurrence-free survival (25.7 months RS vs. 20.0 months OS, *p* = 0.500) were similar between groups. In fact, the only factors which influenced overall survival on multivariable analysis were alfa-fetoprotein (HR 1.43 [1.21–1.69]), size ≥10 cm (HR 1.73 [1.43–2.10]), and macrovascular invasion (HR 1.19 [1.01–1.41]). On subgroup PSM analysis, the “huge” HCC subgroup included 47 RS and 83 OS patients. Shorter operative time (220 min vs. 250 min, *p* = 0.005), lower perioperative blood loss (200 vs. 500 mL, *p* < 0.001), and reduced length of hospital stay (7 days vs. 10 days, *p* < 0.001) were seen in the RS group, whereas complication rate, overall survival, and recurrence-free survival were comparable between groups.

Given that overweight (BMI ≥ 25 kg/m^2^) and obese (≥30 kg/m^2^) individuals comprise a high percentage of the population [28], Lin et al. published a propensity score matched analysis to compare outcomes for overweight individuals undergoing RS vs. OS at a single Chinese institution [29]. Including overweight and obese patients diagnosed between 2010–2020, patients Child-Pugh A/B7 disease who underwent an R0 resection were included. Propensity score matching incorporated demographics, etiology of HCC, laboratory values, cirrhosis/severity, histological features of tumor, and size/number of tumors, and resulted in well-matched groups of 104 patients each. Patients undergoing RS had less estimated blood loss (75 mL vs. 300 mL, *p* < 0.001), were less likely to receive transfusion (9.6% vs. 19.2%, *p* = 0.048), had longer Pringle time (25.5 min vs. 18.0 min, *p* = 0.041), were less likely to suffer minor (Clavien-Dindo < 3) complications (1.9% vs. 8.7%, *p* = 0.030), and had a shorter length of stay (5 days vs. 9 days, *p* < 0.001). Ninety-day mortality (1% RS vs. 0% OS) and major complications (Clavien-Dindo III-V, 1.9% both groups) were statistically similar between groups. For the obese subset, RS demonstrated trends for shorter operative time (135 vs. 204 min, *p* = 0.005), less estimated blood loss (50.0 vs. 350.0 mL, *p*  <  0.001), and shorter length of stay (4 vs. 9 days, *p* < 0.001). Notably, a multivariable analysis was performed to identify factors predictive of increased estimated blood loss; predictive variables included tumors ≥5 cm (OR 2.42 [1.18–4.96]) and operative time ≥180 min (OR 3.692 [1.75–7.80]); protective variables included RS (OR 0.13 [0.06–0.29]) and albumin > 35 g/L (OR 0.24 [0.07–0.77]).

### 3.3. Other Applicable Studies

#### 3.3.1. RS and Margins

Shapera et al. conducted a single-center retrospective analysis investigating the association of HCC RS margins with outcomes [30]. Fifty-eight patients with HCC who underwent RS (2016–2022) were included for analysis and were stratified according to tumor distance to margin (≤1 mm, 1–10 mm, ≥10 mm). Margins ≥ 10 mm were associated with a 5-year survival advantage (88% vs. 47% [1–10 mm] vs. 24% [≤1 mm], *p* = 0.013).

#### 3.3.2. The Impact of Metabolic Syndrome and Obesity

With a decline in hepatitis-C secondary to the introduction of highly effective and non-toxic antiviral agents (introduced around 2011; cure rates ~90%), NASH, or MASH (metabolic dysfunction-associated steatohepatitis), is one of the fastest growing etiologies of HCC [31]. In fact, 25% of the global population is estimated to have nonalcoholic fatty liver disease, and among those with NASH cirrhosis, the incidence of HCC may be as high as 2.6% [32]. These trends underscore the need to understand HCC in the context of metabolic syndrome, characterized by obesity, insulin resistance, dyslipidemia, and hypertension, and to describe the outcomes of RS in this cohort [33].

Rayman et al. published a PSM analysis analyzing cost and short/long-term outcomes in patients with metabolic syndrome (MS) vs. without metabolic syndrome (WO) undergoing RS between 2016–2020 at a single institution in Florida [33]. Matching was performed based on demographics, American Society of Anesthesiologists score, Model for End-Stage Liver Disease/Child-Pugh score, and IWATE score [34]. Seventeen patients were included in each group. Patients were well-matched, aside from an expectedly higher BMI in the MS group (23 kg/m^2^ vs. 32 kg/m^2^, *p* < 0.0001). In-hospital mortality (0% MS vs. 6% WO), post-operative complications (Clavien-Dindo ≥ 3; 0% MS vs. 6% WO), length of stay (5 days MS vs. 6 days WO), and readmissions within 30 days (6% in both groups) were statistically similar between groups. Additionally, no significant differences were noted in overall costs ($32,952 MS vs. $26,890 WO, *p* = 0.68). One, 2, and 3-year overall survivals were similar as well (1-year: 84% MS vs. 77% WO; 2-year: 84% MS vs. 61% WO; 3-year: 45% MS vs. 61% WO, *p* = 0.42).

#### 3.3.3. RS and “Huge” HCC

Wu et al. (2024, China) published their single-surgeon experience of 337 patients who underwent RS, comparing “huge” HCCs (≥10 cm) to smaller HCCs (<10 cm) [35]. After propensity score matching, 21 “huge” HCCs were compared to 63 smaller HCCs. “Huge” HCCs took longer to resect (120 vs. 170 min, *p* = 0.024) and stayed in the hospital for a slightly longer time (6 vs. 5 days, *p* = 0.046), but had similar rates of post-operative morbidity (i.e., postoperative liver failure [n = 0 for both groups], biliary leak [n = 0 for both groups]) and in-hospital mortality (n = 0 for both groups). R0 margin status was achieved 100% of the time for “huge” HCCs vs. 97% of the time for smaller HCCs (*p* = 1.00).

#### 3.3.4. RS vs. Microwave Ablation

Given the applicability of RS for patients with early/small HCCs that traditionally may have been treated with microwave ablation (MWA), Ding et al. compared RS to MWA for patients with early HCC (Barcelona Clinical Cancer Stage 0-A) from 2013–2019 [36]. PSM was performed based on a variety of demographic and clinicopathologic variables, resulting in 122 patients in each group. While 3-year overall survival (91.3% MWA vs. 91.5% RS, *p* = 0.44) and cancer-specific survival (91.3% RS vs. 91.5% MWA, *p* = 0.96) were similar between groups, RS trended towards improved recurrence-free survival at 3 years, though statistical significance was not appreciated (65.8% RS vs. 52.2% MWA, *p* = 0.097). Predictably, RS was associated with a longer operative time (178 vs. 56 min, *p* < 0.001).

## 4. Discussion

Minimally invasive general surgery has been associated with decreased pain, post-operative morbidity, hernia, and length of stay compared to OS across a variety of surgical procedures, including bariatrics, cholecystectomy, appendectomy, small bowel surgery, colorectal surgery, hernia surgery, urological surgery, and gynecological surgery, amongst others [37,38,39,40,41,42,43]. Moreover, minimally invasive surgery conveys similar long-term/oncological outcomes in the context of cancer surgery if similar surgical oncological principals are applied [44,45,46]. Still, liver surgeons have been relatively slower to adapt this method, likely owing to small anatomical workspaces enriched with important vascular structures that are not conducive to limited degrees of freedom/articulation and a 2-D video interface. Fortunately, robotic surgery theoretically overcomes these limitations, as supported by studies suggesting feasibility of RS for pancreaticoduodenectomy [47]; which poses the question: do the benefits of minimally invasive (robotic) surgery translate to liver surgery?

HCC is the most common primary liver cancer and carries a poor 5-year prognosis, though well-designed screening programs have allowed for identification of early disease amendable to resection [1,2,6]. Hence, given the increased availability of robotic liver surgery, more surgeons capable of performing robotic hepatectomy, and early identification of more patients with resectable HCC, a contemporary comparison of RS vs. LS/OS is warranted. Herein, we reviewed pertinent studies from 2018–2024 that specifically evaluated robotic hepatectomy for HCC.

In the studies reviewed, RS yielded comparable long-term and oncological outcomes (i.e., margin status, recurrence-free survival) to LS. While some studies noted higher rate of transfusion with RS compared to LS, others found the opposite. The former authors suggest this difference may be due to the use of bipolar instruments for hepatic parenchymal transection with RS as opposed to CUSA, an ultrasonic energy device, with LS [48]; however, differences in perioperative blood loss between these devices have not been demonstrated in randomized-controlled-trials [49]. The latter authors attribute the lower blood loss with RS to improved dexterity and better tissue handling compared to LS. Similarly, RS yielded comparable short-term, long-term, and oncological outcomes to OS. Many studies highlight the longer operative time but shorter length of stay associated with RS. Interestingly, Zhang et al. found that, when defining RS competence as having passed the learning curve and having performed ≥30 robotic hepatectomies, RS was associated with shorter operative time and lower estimated blood loss compared to OS, emphasizing the impact of learning curve on optimization of outcomes [27]. Compared to OS, RS was generally associated with shorter hospital stays and reduced major complications. These trends, particularly shorter length of stay, are like those observed with robotic surgery for thoracic disease, gastric cancer, colorectal cancer, kidney/bladder cancer, and gynecological disease [50,51,52,53,54,55,56].

Regarding RS vs. LS, it is likely that some inconsistencies noted amongst studies (specifically operative time and blood loss) are due to surgeons being at different stages along the learning curve, which is often a limitation of RS analyses [57,58]. Regarding RS vs. OS, while RS was nearly always associated with longer operative time, as demonstrated by Zhang et al. (2024), this trend may be mitigated by adequate surgeon exposure and training with RS [27]. Similarly, differences in blood loss between RS and OS may be due to operator comfort and learning curve. While we only evaluated one study that showed a difference in overall survival, many studies trended towards improved survival with RS before the 5-year mark. Nonetheless, it is likely that confounding factors may have been responsible for this finding; hence, future studies should attempt to identify and control for these factors. Additionally, RS likely represents a safe alternative to MWA for appropriately selected patients [36]. The studies we have evaluated also suggest that the safety profile of RS likely translates to special populations as well, including those with metabolic syndrome, obesity, and large/“huge” HCCs [27,29,33,35].

In addition to enhanced articulation, tremor reduction, and increased visual acuity, the RS system is compatible with unique energy devices which support safety and efficiency. This is especially critical in hepatic surgery, given the complexity of hepatic parenchymal division. Bipolar vessel sealer technology, including Da Vinci’s SynchroSeal and Vessel Sealer Extend, minimize lateral energy spread while allowing for a wide range of articulation to divide/ligate vessels up to 5–7 mm in diameter [59]. The unique ability in RS to utilize two of these bipolar instruments simultaneously for grasping, retraction, dissection, and hemostasis (i.e., the double bipolar technique) promotes safety and efficiency. Harmonic scalpel devices can deliver significant friction to cut and coagulate tissues, overcoming monopolar/bipolar issues of energy spread, direct, and indirect coupling, and are useful adjunctive instruments compatible with RS [60]. The lack of a robot-compatible CUSA device is a pitfall; however, this can be overcome by utilizing an additional bedside surgeon capable of using the laparoscopic version of the device while RS proceeds from the console.

One of the main criticisms of RS is cost. This contributed, in no doubt, to selection bias in Wang et al., in which patients had to pay approximately $8000 out of pocket for RS (vs. $0 for OS, paid for by national Taiwanese healthcare coverage) [19]. While we did not identify any studies specifically analyzing costs, this issue was investigated by Daskalaki et al. in 2017, who found that RS (while being associated with less estimated blood loss and less morbidity) cost, on average, $36,040, compared to OS, which costed an average of $39,924 [61]. This difference was consistent when considering readmissions (which were similar between groups, 6% RS vs. 9% OS), translating to a cost difference of about $4000 favoring RS. It is possible that, despite a higher intraoperative cost for RS, decreased morbidity and decreased length of stay significantly mitigate the upfront cost, analogous to reports in bariatric surgery [62]. Moreover, one might expect this difference to become more pronounced as more surgeons overcome the RS learning curve.

## 5. Conclusions

In conclusion, the overall findings of the studies we evaluated suggest that RS is a safe alternative to LS and OS that provides similar long-term outcomes, while decreasing post-operative complications and length of stay. As surgeons become more familiar with RS, it may become the least morbid and most cost-effective platform to resect appropriately selected HCCs.

## Figures and Tables

**Table 1 cancers-16-03806-t001:** Studies comparing RS vs. LS.

Author	Year	Study Years	Study Location	n	RS	LS	Study Design	Positive Findings	Outcomes
Duong	2021	2010–2015	USA; National Cancer Database	492	123	369	Retrospective PSM analysis Comparing RS vs. LS Inclusion: Stage I HCC	6.5% RS patients with lymph nodes obtained vs. 2.0% LS patients	5-year overall survival superior for RS (63%) vs. LS (45%)
Lim	2021	2011–2017	Europe; 3 institutions	93	44	49	Retrospective Comparing RS vs. 3-D LS 2 institutions exclusively contributed RS (2011–2017), whereas the other institution exclusively contributed LS (2015–2017)	RS longer operative time (509 min) than LS (394 min) RS less likely to have Pringle maneuver (20% vs. 65%) RS more likely to receive blood transfusion (14% vs. 2%)	3-year overall survival same between RS (91%) vs. LS (82%) 3-year recurrence free survival same between RS (48%) vs. LS (24%)
Liu	2022	2017–2022	China; single institution	131	44	87	Retrospective Comparing RS vs. LS short-term outcomes	RS longer operative time (55 min) than LS (207 min) RS less intraoperative blood loss (119 mL) vs. LS (197 mL)	No 30-day or 90-day mortality in either group

LS = Laparoscopic surgery; PSM = Propensity score matching; RS = Robotic surgery.

**Table 2 cancers-16-03806-t002:** Studies comparing RS vs. OS.

Author	Year	Study Years	Study Location	n	RS	OS	Study Design	Positive Findings	Outcomes
Wang	2018	2013–2016	Taiwan; single institution	240	63	177	Retrospective Comparing RS vs. OS Inclusion (only applied to RS): ○ Tumor < 5 cm ○ Peripheral tumors, segments II-VI ○ No prior foregut surgery Patients paid out of pocket for RS, whereas national insurance covered OS (+$8000)	RS tumors smaller (3.1 cm vs. 3.6 cm) RS more likely to undergo wedge as opposed to anatomical resection (67% vs. 45%) RS longer operative time (296 min vs. 182 min) RS shorter hospital stay (6.2 days vs. 8.2 days)	3-year overall survival same between RS (97.7%) vs. OS (92.3%) 3-year disease-free survival same between RS (71.9%) vs. OS (61.6%)
Pesi	2021	N/a	Europe; 2 institutions	54	23	31	Retrospective Comparing RS vs. OS	RS less intraoperative blood loss (186 mL vs. 354 mL)	3-year overall survival rate same between RS (87%) vs. OS (78%) 3-year disease-free survival rate same between RS (54%) vs. OS (37%)
Di Benedetto	2022	2010–2020	Europe; USA; 5 institutions	212	106	106	Retrospective PSM analysis Comparing RS vs. OS short-term outcomes	RS longer operative time (295 min vs. 200 min) RS shorter hospital stay (4 days vs. 10 days) RS higher intraoperative blood loss (200 mL vs. 100 mL) RS less major complications (2.8% vs. 11.3%) and less post-operative hepatic failure (28.3% vs. 7.5%)	90-day overall survival similar between RS (99.1%) and OS (97.1%)
Zhang	2022	2010–2020	China; single institution	278	100	178	Retrospective PSM analysis Comparing RS vs. OS for elderly patients Inclusion: ○ ≥65 years old ○ Child-Pugh A/B7 ○ R0 resection Exclusion ○ Pre-operative antitumor treatment	RS lower intraoperative blood loss (150 mL vs. 200 mL) RS lower post-operative complications (7% vs. 17%) RS shorter length of stay (6 days vs. 9 days)	90-day overall survival similar between RS (99%) and OS (99%) 5-year overall survival similar between RS (43%) vs. OS (49%) 5-year recurrence-free survival similar between RS (21%) vs. OS (25%)
Zhu	2023	2015-N/a	China; single institution	168	56	56; also 56 for LS	Prospectively built cohorts PSM analysis Comparing RS vs. LS vs. OS Inclusion: Barcelona stage 0-A HCC	RS and LS with longer operative time than OS (220 min vs. 210 min vs. 158 min, resp.) Length of stay shorter for RS and LS (8 and 6 days, resp.) than OS (12 days)	5-year overall survival similar between groups (74.4% RS, 76.8% LS, 78.6% OS) 5-year disease-free survival similar between groups (51.8% RS, 51.3% LS, 57.9%)
Zhang	2024	2010–2020	China; 8 institutions	745	280	465	Retrospective PSM analysis Comparing RS to OS for large HCCs (5–10 cm)	RS shorter operative time (181 min vs. 201 min) RS lower intraoperative blood loss (200 mL vs. 400 mL) RS shorter hospital stay (3 vs. 9 days) RS less major complications (2% vs. 6%)	Median overall survival similar between RS and OS (68.9 vs. 64.4 months, resp.) Median recurrence-free survival similar between RS and OS (25.7 vs. 20 months, resp.)
Lin	2023	2010–2020	Cina; single institution	208	104	104	Retrospective PSM analysis Comparing RS to OS short-term outcomes for overweight individuals with HCC Inclusion: ○ Child-Pugh A/B7 ○ BMI ≥ 25 ○ R0 resection	RS lower intraoperative blood loss (75 mL vs. 300 mL) RS less likely to receive transfusion (9.6% vs. 19.2%) RS less likely to suffer minor complications (1.9% vs. 8.7%) RS shorter length of stay (5 vs. 9 days)	90-day mortality similar between groups (1% RS vs. 0% OS)

BMI = Body mass index (kg/m^2^); OS = Open surgery; PSM = Propensity score matching; RS = Robotic surgery.

**Table 3 cancers-16-03806-t003:** Study comparing robotic surgery (RS) vs. microwave ablation (MWA).

Author	Year	Study Years	Study Location	n	RS	MWA	Study Design	Positive Findings	Outcomes
Ding	2021	2013–2019	China; single institution	244	122	122	Retrospective PSM analysis Comparing RS vs. MWA Inclusion: ○ Pathological confirmation of HCC ○ Barcelona Clinic Liver Cancer stage 0-A ○ No treatment history ○ >6 months follow-up Exclusion: ○ Combined MWA or RFA during RH ○ Palliative treatment ○ Intra-operative conversion from RS to open	RS longer operative time (178 vs. 56 min)	3-year overall survival same between RS (91.3%) vs. MWA (91.5%) 3-year cancer-specific survival same between RS (91.3%) vs. MWA (91.5%) 3-year recurrence-free survival same between RS (65.8%) vs. MWA (52.2%)

MWA = Microwave ablation; PSM = Propensity score matching; RS = Robotic surger.

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
