# Peer review of "A Contemporary Review of Robotic Resection for Hepatocellular Carcinoma"

_cancers, 2024, doi:10.3390/cancers16223806_

Round 1
Reviewer 1 Report
Comments and Suggestions for Authors
This study was well-written and full of information. More information about robotic hepatectomy margins, metabolic syndrome, huge HCCs, and comparson with microwave ablation were reported that made the paper interesting.
But the maniulation of energy device could be put in the discussion, such as vessel extend and Synchroseal, Harmonic Scalpel , lack of CUSA and dual bipolart tchnique. The concept of bedside and console surgeon is another important issue for safety and effective practice.
Reviewer 2 Report
Comments and Suggestions for Authors
This is a review of articles published from 2018-2024 comparing robotic liver resections for HCC with laparoscopic and open surgery, including 1 comparing with microwave therapy. Whilst comparing robotic surgery to laparoscopic or open does seem reasonable, it would have been clearer to readers if the single paper comparing robotic surgery to MWT is left out.
The intent of the paper was stated as a comparison of robotic liver resection to laparoscopic /open resections. It is therefore somewhat out of place for the comparison of robotic surgery for those with and without metabolic disorders (section 3.3), without any comment on how this compares to that for laparoscopic/open surgery in similar clinical context.
Similarly, the question of margins is also addressed only for robotic surgery without comparisons to laparoscopic/open resections (section 3.3).
Comments on the Quality of English LanguageSome typographical and grammatical errors throughout the text.
Writing style could be improved. The authors seem to lead in each new paragraph with the year of publication of papers reviewed (eg line 107 - "In 2021, ...", line 135 - "In 2022,...", line 167 - "In 2021,...", line 182 - "In 2022,...", line 198 - "In 2022,...", line 212 - "In 2023,...", line 223 - "In 2024,...", line 270 - "In 2022,...", line 284 - "In 2022,...", and line 308 - "In 2022,...", which makes reading less interesting.
Round 2
Reviewer 2 Report
Comments and Suggestions for Authors
Noted the authors comments and reply to my previous review. Although the authors have chosen to keep the paper on RS vs MWT and also the section on RS for metabolic and other disorders, I can understand their intent and am ok with that. The language has been improved to make reading easier.